# Individual, Family, and Socioeconomic Contributors to Dental Caries in Children from Low- and Middle-Income Countries

**DOI:** 10.3390/ijerph19127114

**Published:** 2022-06-10

**Authors:** Madiha Yousaf, Tahir Aslam, Sidra Saeed, Azza Sarfraz, Zouina Sarfraz, Ivan Cherrez-Ojeda

**Affiliations:** 1Research and Publications, Fatima Jinnah Medical University, Queen’s Road, Lahore 54000, Pakistan; madiyousaf08@gmail.com; 2Research, Central Park Medical College, Lahore 54600, Pakistan; tahiraslam398@gmail.com; 3Research, Gujranwala Medical College, Gujranwala 52250, Pakistan; sdra.saeed@gmail.com; 4Pediatrics and Child Health, Aga Khan University, Karachi 74000, Pakistan; azza.sarfraz@aku.edu; 5Allergy and Pulmonology, Universidad Espíritu Santo, Km. 2.5 Vía La Puntilla, Samborondón 0901-952, Ecuador; 6Respiralab Research Center, Guayaquil 0901-952, Ecuador

**Keywords:** dental caries, risk factors, breastfeeding, bottle-feeding, fluoride, socioeconomic status, education, low- and middle-income countries

## Abstract

Introduction: Collective evidence on risk factors for dental caries remains elusive in low- and middle-income countries (LMICs). The objective was to conduct a systematic review and meta-analysis on risk factors for dental caries in deciduous or permanent teeth in LMICs. Methods: Studies were identified electronically through databases, including Cochrane Oral Health Group Trials Register, Cochrane Central Register of Controlled Trials, PubMed/MEDLINE, and CINAHL, using “prevalence, dental caries, child, family, socioeconomic, and LMIC” as the keywords. A total of 11 studies fit the inclusion criteria. Quality assessment of the included studies was performed using the Newcastle-Ottawa Scale (NOS). The MedCalc software and Review Manager 5.4.1 were used. Results: From 11,115 participants, 38.7% (95% CI: 28.4–49.5%) had caries and 49.68% were female. Among those with caries, 69.74% consumed sugary drinks/sweets (95% CI: 47.84–87.73%) and 56.87% (95% CI: 35.39–77.08%) had good brushing habits. Sugary drinks had a two times higher likelihood of leading to caries (OR: 2.04, *p* < 0.001). Good oral hygiene reduced the risk of caries by 35% (OR: 0.65, *p* < 0.001). Concerning maternal education, only secondary education reduced the likelihood of caries (OR: 0.96), but primary education incurred 25% higher risks (OR: 1.25, *p* = 0.03). A 65% reduction was computed when caregivers helped children with tooth brushing (OR: 0.35, *p* = 0.04). Most families had a low socioeconomic status (SES) (35.9%, 95% CI: 16.73–57.79), which increased the odds of caries by 52% (OR: 1.52, *p* < 0.001); a high SES had a 3% higher chance of caries. In the entire sample, 44.44% (95% CI: 27.73–61.82%) of individuals had access to dental services or had visited a dental service provider. Conclusion: Our findings demonstrate that high sugar consumption, low maternal education, and low and high socioeconomic status (SES) increased the risk of dental caries in LMICs. Good brushing habits, higher maternal education, help with tooth brushing, and middle SES provided protection against caries across LMIC children. Limiting sugars, improving oral health education, incorporating national fluoride exposure programs, and accounting for sociodemographic limitations are essential for reducing the prevalence of dental caries in these settings.

## 1. Introduction

Dental caries is a global public health challenge, specifically in low- and middle-income countries (LMICs), and it has been linked to underlying socioeconomic and social disadvantages [1]. Data on the rising prevalence of caries are mainly present from uncoordinated independent studies in 88 countries (~45%) globally [2]. The trends are skewed as the prevalence is 7–70 fold higher in LMICs, with up to a 70% prevalence compared to 1–12% in high-income countries (HICs) [3]. The World Health Organization (WHO) data suggest that the decline in the prevalence of caries in the first decade of this century is only noted in HICs, e.g., USA and Western Europe, whereas LMICs have either had a less explicit reduction or an increase in burden in certain countries, such as Gambia, Croatia, Saudi Arabia, and Moldova [4]. Regional differences in oral health disparities for children are also noted in quality of care, access to care, cost of care, oral health education, and health literacy [5,6,7]. The majority of health service expenditures in HICs is invested in preventive oral care, whereas in LMICS, it is used for emergency oral care and pain relief [8]. 

Caries has adverse effects on both oral and general health, as noted by the dental community and public health authorities [9]. Caries in permanent and deciduous teeth is well recognized as a multifactorial disease. At present, many independent risk factors are being cited across LMICs in the literature, including individual (diet [10], oral hygiene [11], feeding practices, low birth weight [12], hereditary enamel defects [13]), family (maternal education [14], oral health knowledge [15], attitude and practice [16], household income level [17]), sociocultural (area of residence and cultural values [18]), environmental (access to fluoridated water [19]), and economic (public health policy and investment [20]) factors. It is necessary to establish important prophylactic measures, such as developing regular oral hygiene, e.g., proper brushing and flossing methods, fluoride application, minimizing sugar intake, and improvement of health education among primary caregivers [21,22]. 

Significant gaps exist in the literature on the primary risk factors and their degree of association with caries, as they vary among children across different settings. Earlier, poor oral hygiene and dietary habits were the primary risk factors in LMICs [23]. However, sociodemographic inequality is increasingly recognized as the primary risk factor for caries in the same populations [23,24]. Recent studies have suggested a bi-directional association of caries with stunting as an independent risk factor in LMIC settings, where undernutrition is highly prevalent [25]. Undernutrition, however, is also multifactorial. Underlying poverty, inadequate birth spacing, unclean water, low-quality food, an unhealthy environment, and a poor access to health facilities are implicated in high stunting rates in LMICs [26,27]. As a predictive indicator of dental caries, undernutrition does not focus on more upstream contributors to dental caries [28]. If the risk factors for caries development are identified and timely prevented in LMIC children, a multiplier effect could be achieved. 

Dental caries requires rigorous assessment of caries risk and carious lesion activity for focused treatment plans, which may vary depending on different dental practices [29]. For instance, zirconia crowns (ZCs) are being used due to their acceptable efficacy and aesthetic acceptance compared to the traditionally used stainless steel crowns (SSCs) for primary teeth restoration [30]. Carious teeth treatment is evolving, albeit predominantly in HICs, and it is necessary to improve dental care in children [31]. While data are emerging from HICs, there are gaps in the collective evidence on the risk factors known to cause caries in LMICs that require further exploration [32]. The aim of this research was to systematically assess the associations of individual, maternal, and socioeconomic risk factors on dental caries in children from LMICs.

## 2. Methodology

A systematic review and meta-analysis were conducted and reported, adhering to the Preferred Reporting Items for Systematic Reviews and Meta-Analyses (PRISMA) 2020 statement guidelines [33]. Dental caries in the deciduous and/or permanent dentition was considered and there was no limit to the children’s age. The primary objective of this study was to assess the prevalence of child, maternal, and socioeconomic factors pertaining to the presence of dental caries in children with deciduous or permanent dentition. The dmft (decayed, missing, and filled primary teeth) and DMFT (decayed, missing, and filled permanent teeth) indices were used as a global measure for the assessment of oral health via a categorization of the number of decayed teeth, the number of treated teeth, and the number of teeth missing due to decay [34]. The secondary objective was to assess the correlation and causation (i.e., meta-analytical technique) of the various child, maternal, and socioeconomic factors to the presence or absence of dental caries in the children. PROSPERO 2022 CRD42022331041.

### 2.1. Study Variables

The exposure variables were child, family, and socioeconomic factors, reported as prevalence. The outcome variable was dental caries, including absence and presence in primary or permanent dentitions. The studies were included if they contained measures of child-level, family-level, and socioeconomic-level factors contributing to dental caries in children residing in low- and middle-income countries (LMICs), as defined by the World Bank [35]. 

### 2.2. Search Strategy

Searches were conducted in March 2022. The databases searched included the Cochrane Oral Health Group Trials Register (11 March 2022), the Cochrane Central Register of Controlled Trials (CENTRAL; 11 March 2022), MEDLINE (1946 to 11 March 2022), and CINAHL via EBSCO (1937 to 11 March 2022). The following MESH keywords were used: “Prevalence”, “Dental Caries”, “Child”, “Family”, “Socioeconomic”, and “LMIC”, applying the BOOLEAN logic (And/Or). Hand searches of citation lists of identified reviews and expert consultations were conducted to determine further studies. Abstracts and unpublished studies were not included. No language restrictions were placed in our search. After any duplicate citations were removed, the electronic searches retrieved 3405 references. The search strategy is presented in Figure 1. 

### 2.3. Study Selection

The retrieved records (titles and abstracts) were screened independently by two reviewers in order to clarify the inclusion criteria (A.S., Z.S.), with a third reviewer present for any disagreements (I.C.-O.). The full texts underwent independent duplicate screening. The evidence was grouped into the following three characteristics, which were identified as contributing factors: child-level, family-level, and socioeconomic level for quantitative and qualitative data synthesis using the “best available evidence” [36]. 

### 2.4. Data Extraction

Data extraction and quality appraisal were undertaken by two reviewers and checked by a third reviewer. Evidence synthesis was conducted by using a vote counting method that is suited to data from a heterogeneous group of studies [37] and by weighing the evidence showing a positive relationship between exposure (factors) and outcome (dental caries), along with showing a negative association based on the direction of the effect. The data were extracted into a shared spreadsheet, where the characteristics of the included studies were listed as follows: author–year, type of study, age (years) at enrollment, duration of follow-up (years), country, sample size (N), type of dentition, measurements taken, any dental caries (n, %), dmft/DMFT scores (mean, SD), anthropometric status (n, %), and primary objective. The child-level characteristics were entered as follows: gender (n, %), ever breastfed (n, %), ever bottle-fed (n, %), bottle-fed at night (n, %), nutritional patterns (n, %), and brushing patterns/oral hygiene status (n, %). The family-level characteristics were grouped as follows: maternal age (years), parental educational status (n, %), help with child tooth brushing (n, %), and fluoride toothpaste (n, %). Finally, the socioeconomic-level characteristics were noted under socioeconomic status (SES) (n, %), setting (n, %), and access to/visit to dental services (n, %). Data were formulated into proportion plots to depict the study characteristics and the weight of evidence in relation to the review question. This was supplemented with meta-analytical presentations of odds ratio (OR) forest plots of selected outcomes along with a narrative synthesis of findings.

### 2.5. Data Analysis

The pooled estimate of dental caries in the included sample from LMICs was calculated with a 95% confidence interval and the data were displayed with a random-effects model. The random-effects model, as applied to this meta-analysis, was considered more appropriate for the current study due to the non-randomized nature of the included studies. In the case that a substantial heterogeneity is present among the included studies, a random-effects model weighs the study more equally and is considered generally acceptable. The Cochran’s Q test and the I^2^ index were both used to present a variance between studies and heterogeneity estimates. The two tests were reported as percentages. In the case that the I^2^ index was >75%, it indicated a high heterogeneity, whilst values of 30–70% established a moderate heterogeneity and <25% values established a low heterogeneity. Forest plots were presented to show prevalence estimates of child, maternal, and socioeconomic trends among children with carries, using 95% CI. The analysis was conducted using a MedCalc statistical software (V 19.5.3). The odds ratio (OR) was computed using a Review Manager (RevMan 5.4.1), whereby a random-effects model was used along with 95% CI. The results were computed by comparing children with carries to those without carries. The findings were reported as OR, 95% CI, I^2^ index, and level of significance (*p* value).

### 2.6. Risk of Bias Assessment

The Newcastle-Ottawa Scale (NOS) was used for assessing the quality of non-randomized studies in this meta-analysis. The scale contains 8 items in 3 domains and has a maximum score of 9. Studies that scored from 7–9 are considered high quality; studies with a score of 4–6 are considered high risk; studies with a score of 0–3 are considered to have a very high risk of bias.

## 3. Results

Figure 1 presents a PRISMA flowchart. We identified 9372 records, out of which 5967 were duplicates. A total of 3405 records were screened for titles and abstracts, of which we retrieved 1939 records. Out of these, 1886 records were irrelevant to our study’s outcomes, 53 records were assessed for eligibility and 11 articles were eligible for inclusion (42 studies were ineligible for the reasons attached in Figure 1). The studies were assessed for quality, wherein 11 studies were included in the qualitative synthesis and all 11 studies pertaining to the contributing factors were included in the quantitative synthesis. A total of 11 articles formed the bases of this systematic review; the inter-reader κ agreement was 0.88 ± 0.05.

### 3.1. Study Characteristics

Table 1 summarizes the baseline characteristics of the 11 included studies. All of the included studies reported primary or secondary observational data from 2001 to 2022. The pooled proportion of caries among the 11 included studies in the total sample of 11,115 individuals was 38.7% (95% CI = 28.4% to 49.5%) (Q = 1277.61; DF = 10; *p* < 0.0001; I^2^ = 99.22%; Egger’s test = 3.83; Kendall’s Tau = 0.09) (Figure 2). The studies were conducted across six different LMICs, including Brazil [38,39,40,41,42], Nigeria [43,44], Cambodia [45], Tanzania [46], Mexico [47], and Ethiopia [48]. All the studies used either the dmft (decayed, missing, and filled primary teeth) or DMFT (decayed, missing, and filled permanent teeth) index, except for Pérez et al. [47], who used the International Caries Detection and Assessment System II (ICDAS II) 1–6 index. The primary objective of the six studies [39,41,42,46,47,48] was to assess the contributing factors of dental caries, whereas five studies evaluated them as secondary objectives [38,40,43,44,45]. Four studies reported cross-sectional data of prospective cohorts, including birth cohorts, followed for three years [40] and twelve years [38], respectively, and childhood cohorts followed for one year each [41,45]. Seven studies [39,40,41,42,43,45,46] reported factors associated with primary dentition. The individual proportions, 95% CI, and weights are enlisted in Appendix A.

### 3.2. Child-Level Characteristics

Table 2 summarizes the child-level characteristics for children with and without caries, including gender, breastfed status, bottle-fed patterns, nutritional patterns, brushing patterns, and oral hygiene status. 

#### 3.2.1. Gender

The pooled proportion of female gender (n = 993/1999) in the population with caries was 49.68% (95% CI = 47.49 to 51.86) (Q = 4.4274; DF = 5; *p* = 0.4897; I^2^ = 0% (Figure 2).

#### 3.2.2. Breastfed/Bottle-Fed Patterns

Only one study [39] reported breastfeeding patterns; 17.4% of 907 children with caries were breastfed, whereas 19.3% of 689 children without caries were breastfed (*p* < 0.01).

#### 3.2.3. Nutritional Patterns

The pooled proportion of sugary drinks/sweets consumed in the population with caries was 69.74% (95% CI = 47.84 to 87.73) (Q = 470.84; DF = 6; *p* < 0.0001; I^2^ = 98.73%) (Figure 2). There was a 2.04 times higher likelihood of caries among participants consuming sugary drinks/sweets compared to the non-caries group (OR = 2.04, 95% CI = 1.59 to 2.6; *p* < 0.001, I^2^ = 28%) (Figure 2).

#### 3.2.4. Oral Hygiene Patterns

The pooled proportion of good brushing habits/≥ 2 daily teeth brushing in the population with caries was 56.87% (95% CI = 35.39 to 77.08) (Q = 643.04; DF = 6; *p* < 0.0001; I^2^ = 99.07%) and there was a 35% reduction in acquiring caries (OR = 0.65, 95% CI= 0.46 to 0.92; *p* = 0.01, I^2^ = 81%) (Figure 2). 

### 3.3. Family-Level Characteristics

Table 3 collates family-level characteristics of children with and without caries including maternal age, parental education status, help with child tooth brushing, and fluoride toothpaste use.

#### 3.3.1. Maternal Age

The maternal age was reported in two studies [39,41]. Saraiva et al. [39] provided maternal age at the time of child birth in the caries and non-caries group. In the group of children with caries, 138 (15.2%) of the mothers were aged <20 years, 201 (22.2%) were aged 20–29 years, and 141 (15.5%) were aged >29 years; in the non-caries group, 147 (21.4%) were aged <20 years, 114 (16.6%) were aged 20–29 years, and 74 (10.7%) were aged >29 years (*p* < 0.01). Fraiz et al. [41] reported that the mothers’ age was 26.3 (SD = 5.1) years when the children were born, with no significant differences between caries and non-caries groups.

#### 3.3.2. Maternal Education

The pooled prevalence of no formal education among the mothers of children with caries was 10.14% (95% CI = 3.34 to 20.07) (Q = 55.09; DF = 2; *p* < 0.0001; I^2^ = 96.37%) (Figure 3). The pooled prevalence of primary education among the mothers of children with caries was 42.46% (95% CI = 27.99 to 57.63) (Q = 433.15; DF = 6; *p* < 0.0001; I^2^ = 98.61%) (Figure 3). The pooled prevalence of secondary and/or higher among the mothers of children with caries was 44.96% (95% CI = 33 to 57.22) (Q = 280.78; DF = 6; *p* < 0.0001; I^2^ = 97.86%) (Figure 3). On computing the odds of children acquiring caries in relation to the mothers’ education, the only indicator that reduced the chance was secondary and higher education status by 4% (OR = 0.96, 95% CI = 0.64 to 1.44; *p* = 0.84, I^2^ = 92%). Mothers who had only primary education incurred a 25% higher chance of the child having caries (OR = 1.25, 95% CI = 1.02 to 1.54; *p* = 0.03, I^2^ = 83%). Mothers who had no education bore a 34% higher change of the child having caries (OR = 1.34, 95% CI = 0.76 to 2.36; *p* = 0.32, I^2^ = 79%) (Figure 3). 

#### 3.3.3. Help with Tooth Brushing

The pooled prevalence of caregivers helping with tooth brushing was 54.08% among children with caries (95% CI = 26.63 to 80.25) (Q = 132.06; DF = 2; *p* < 0.0001; I^2^ = 98.49%) (Figure 4). A 65% reduction was noted when caregivers provided help with tooth brushing (OR = 0.35, 95% CI = 0.13 to 0.93; *p* = 0.04, I^2^ = 95%) (Figure 5). 

#### 3.3.4. Use of Fluoride Toothpaste

The pooled prevalence of children with caries using fluoride toothpaste was 39.1% (95% CI = 16.58 to 64.4) (Q = 72.19; DF = 1; *p* < 0.0001; I^2^ = 98.61%) (Figure 4). 

### 3.4. Socioeconomic Level Characteristics 

Table 4 summarizes the socioeconomic characteristics of children with and without caries including the number of people in the household, socioeconomic status, setting, and access to dental services. 

#### 3.4.1. Number of People in the Household

Ndekero et al. [46] stated that 321 (86.3%) of the children with caries had siblings, whereas 403 (87.8%) of the children without caries had siblings present. Saraiva et al. [39] only included singletons (100%) in their study.

#### 3.4.2. Socioeconomic Status

The pooled prevalence of low socioeconomic status (SES) among children with caries was 35.9% (95% CI = 16.73 to 57.79) (Q = 968.43; DF = 3; *p* < 0.0001; I^2^ = 99.38%) (Figure 4). The pooled prevalence of middle SES among children with caries was 35.34% (95% CI = 18.04 to 55) (Q = 776.64.43; DF = 6; *p* < 0.0001; I^2^ = 99.23%) (Figure 4). The pooled prevalence of high SES among children with caries was 24.51% (95% CI = 16.21 to 33.9) (Q = 82.27; DF = 4; *p* < 0.0001; I^2^ = 95.14%) (Figure 4). Among the children who belonged to the low socioeconomic status group, there was a 52% higher chance of acquiring caries (OR = 1.52, 95% CI = 1.22 to 1.89; *p* = 0.0002, I^2^ = 58%) (Figure 5). Children belonging to the middle socioeconomic status had a 20% less chance of acquiring caries (OR = 0.8, 95% CI = 0.59 to 1.1; *p* = 0.17, I^2^ = 84%) (Figure 5). Children of the high socioeconomic status had a cumulative 3% higher chance of getting caries (OR = 1.03, 95% CI = 0.81 to 1.31; *p* = 0.81, I^2^ = 53%) (Figure 5).

#### 3.4.3. Setting

Two of the studies from Brazil collected data from clinics [40,41], one from public preschools [42], and one was in an urban setting [38]. For the two Nigerian studies, one was conducted in peri-urban households [43] and one was conducted at both public (561 children, 64.3%) and private schools (312 children, 35.7%) [44]. The Cambodian study was conducted in both rural (988 children, 75.6%) and urban (319 children, 24.4%) settings [45]. The Tanzanian study was conducted in rural (661 children, 79.5%) and semi-rural (170 children, 20.5%) settings [46]. Ayele et al. [48] conducted their study in rural and urban communities.

#### 3.4.4. Access to Dental Services

The pooled prevalence of access/visit to dental service providers among both the caries and non-caries groups was 44.44% (95% CI = 27.73 to 61.82) (Q = 667.81; DF = 4; *p* < 0.0001; I^2^ = 99.4%) (Figure 4).

## 4. Discussion

We conducted a systematic review and meta-analysis to explore the prevalence and associated factors of dental caries in children residing in LMICs. Our findings highlight the influence of child–family–socioeconomic factors on the prevalence and risk of developing dental caries. While we cannot confirm the direct association of these factors with caries, our findings suggest a cumulative impact on caries. Dental caries is the highest cause of morbidity in children, with 64.6 million and 62.9 million prevalent cases of caries in permanent and deciduous teeth globally [49]. 

No clear trends emerged concerning breastfed or bottle-fed children in our findings due to the limited data; however, the interplay between breastfeeding and bottle-feeding practices is a crucial component contributing to oral health disparities during the first two years of life [50]. The WHO and UNICEF recommend exclusive breastfeeding for the first six months of age and complementary feeding for the next two years [51]. Avila et al. reviewed seven studies and indicated that breastfed children were less likely to develop dental caries (OR: 0.43; 95% CI: 0.23–0.80) [52]. However, other studies have reported ambiguous results concerning child-feeding habits and dental caries due to the time-dependent compounding of breastfeeding, formula feeding, nocturnal feeding, and other foods/drinks [53,54,55]. In the context of LMICs, higher wealth quintiles are less likely to continue breastfeeding, exclusive or otherwise, and are more likely to formula feed [56]. The drivers for this transition are related to a lack of relevant policies to protect breastfeeding, which are different for richer and poorer families. For wealthier families in LMICs, these include the ability to afford formula and non-human milk, income growth, a feminized workforce, and intense formula marketing [57,58]. However, there is a lack of appropriate policies and guidelines for socioeconomically disadvantaged women in and across LMICs. They are unable to breastfeed their children due to work commitments or cultural misconceptions. Further investigation is needed to carefully control the confounding factors (e.g., the timing of introduction, sugar content of other foods/drinks, and oral hygiene) to understand the association between dental caries and breastfeeding/bottle feeding practices within the different socioeconomic classes in LMIC settings.

The most frequently reported poverty index relevant in LMICs for childhood caries is dietary intake. Our research identifies a measurable problem of high sugar consumption (~69.7% of children with caries) and twice the risk of developing dental caries in our sample (*p* < 0.0001). The consumption of sugar-sweetened beverages (SSBs) and sugary foods has increased globally due to higher affordability in the last two decades [59]. Energy-dense foods are generally more palatable and are available at a lower marginal cost than healthier alternatives in resource-constrained settings [60]. There is a dearth of research promoting dietary interventions in dental practice. Still, even if the optimal threshold of <5% free sugar consumption endorsed by the World Health Organization (WHO) is achieved, there is a risk of dental caries [61]. Educational strategies focusing on improving nutrient quality by maintaining food costs are needed to subsidize healthier culturally relevant recipe ingredients and minimize in-school sales of SSBs and sugary snacks [62]. Similarly, as children start weaning, parental dietary practices are a major influence since healthy eating behaviors are emulated by children [63]. 

The prevalence of dental caries was comparable across lower (35.9%) and middle (35.3%) socioeconomic quartiles (*p* < 0.001). Children belonging to a low SES had the highest increased risk (~54% increased risk, *p* < 0.001), followed by a high SES (~3% increased risk, *p* = 0.81), whereas a middle SES had a 20% lesser chance (*p* = 0.17). The SES in our data was focused on household income, which served as an economic indicator. Many socioeconomic determinants, including health inequality, income, ability to pay for services, and physical and geographical access to dental care services, have directly or indirectly correlated with dental health disparities [64]. We found that a significant number of children (~44%) with dental caries had access to dental services, which suggests the role of numerous socioeconomic constraints beyond geographical limitations in LMICs. 

Our findings build on empirical evidence of socioeconomic inequality in dental health panning across low and middle socioeconomic quartiles across LMICs. Schendicke et al. conducted a global systematic review and meta-analysis of 159 countries and found a higher risk of caries development with low SES in HICs than in LMICs [65]. However, our novel findings of middle SES being mildly protective against acquiring dental caries in LMICs may be due to a fine balance of sociodemographic drivers (e.g., moderate access and affordability to dental care), genetic factors (e.g., low burden of undernutrition leading to fewer enamel defects and less delayed tooth eruption [66]), and good dietary practices (e.g., affordability and access to quality food, higher maternal education, and moderate income) in favor of good oral health [67,68]. 

Nearly 2/5th of children using fluoride toothpaste developed caries in our sample (*p* < 0.0001). Fluoride can improve dental health topically, e.g., toothpaste, mouth rinse, topical treatments in dental clinics, and systemically, e.g., water fluoridation, salt, fluoride supplementation. Both types of applications are known to improve oral health, yet topical application is the most effective [69]. While toothpaste containing fluoride offers protection, the response is dose-dependent [70] and does not eliminate the risk when combined with high sugary diets [71]. Fluoride has historically been a breakthrough in public health for caries prevention through the controlled addition of fluoridated water supplies and the availability of fluoride-containing toothpaste [72]. However, a study from Brazil highlighted the discrepancy of fluoridated tap water only in better-off towns within the country [73]. Countries that have not yet implemented fluoridated programs are primarily from LMICs and require technical assistance and guidance to execute population-wide automated measures [74]. National programs endorsing equitability in fluoridated water, salt, and milk may serve as a practical public health measure against dental caries [75]. Good oral hygiene practices are also pertinent, as seen in our findings of maternal/caregiver tooth brushing (~65% risk reduction) help and tooth brushing ≥ 2 daily (~35% risk reduction).

Our findings confirm an inverse linear correlation between maternal education and the risk of acquiring dental caries, which differs from previous findings of no correlation [76,77]. Mothers with secondary and higher education conferred minor protection (~4% reduced risk, *p* = 0.84), whilst mothers with primary education incurred 25% increased risk (*p* = 0.03) and illiterate mothers attributed the highest risk (~34% increased risk, *p* = 0.32). We confirm a significant correlation between maternal education and reduced dental caries through better oral health beliefs, habits, practices, and behaviors in their children as primary caregivers in LMICs [78]. Maternal education ties in closely with attitude, perception, and family environmental influence on children’s oral health practices [79]. For instance, parents with poor oral health are more likely to have children with dental caries, perhaps due to the interaction between genetic and environmental exposures [80]. Maternal age has been found to have a u-shaped relationship, e.g., mothers under 25 years or over 34 years of age at the birth of a child, with dental caries among children [81]. The mechanism between maternal age and the effects on dental caries is likely due to the different underlying factors among older and younger mother–social factors, baseline health, and health behaviors [81]. We did not find enough data to identify maternal age and correlate it with dental caries prevalence and risk.

### Limitations and Recommendations

In the present study, our outcome was dichotomous, thus, it did not provide data on the severity of the disease. Certain flaws in the sampling technique and sample size were existent due to the methodological flaws of the studies. Another possible limitation was observed because the studies were only analyzed from LMICs; data from non-LMICs may report other contributors of dental caries that were beyond the scope of this review. Our systematic review and meta-analysis studies were not of the highest quality, including cohorts and cross-sectional data. Another important limitation was that we obtained data from six countries (Mexico, Brazil, Cambodia, Tanzania, Nigeria, and Ethiopia) across Latin America, Asia, and Africa. Therefore, we do not expect our findings to be generalizable across LMICs, particularly in other regions. Overall, there is a need for national- and international-level population-based studies with equal representation and socioeconomic representation from rural and urban areas. 

We recommend more longitudinal studies (e.g., mixed-method) that collect data on the link between early tooth brushing, dietary practices, and socioeconomic determinants, including income, education, and access to dental services. Globally, the risk factors for childhood caries that have been recognized as essential may not account for LMIC populations. The primary risk factors that are prominent globally include a high intake of free sugars, poor oral hygiene, and inadequate use of fluoride. However, families are the primary source of health communication about oral health. The combined efforts at individual, family, and community levels are likely to be effective, as observed by Albino et al. [82]. Health promotion may occur through a range of public health interventions (e.g., pregnancy [83], mass communication [84], home visits/telephonic contacts [85], in schools [86]) targeted in low-resource communities. Importantly, as supported by our findings, preventing dental caries requires addressing social and economic challenges. Universal health coverage for all people to receive oral health care, including health promotion, prevention, and treatment, is vital [87]. Even if we use a more conservative approach, identification of dietary trends, family health awareness and practices, community-level influences, and fluoride exposure are essential as preventative measures [88]. Such an approach requires early interventions (e.g., the first year of life), evidence-based interventions (e.g., promotion of good dental practices as behavioral interventions), and risk-based interventions (e.g., high-risk sociodemographic groups), which are pertinent as the preventative measures supported by our findings [89,90].

## 5. Conclusions

This review has highlighted that the balance of best available evidence suggests an association between dental caries and child–family–socioeconomic factors in LMIC settings. Data from clinical studies in LMICs suggest a cumulative causal effect with many explored factors. There is a need for more well-designed prospective studies on the prevention of dental caries through modifying these factors. In addition to obtaining more data on the impact of clinical interventions on dental caries, prospective longitudinal studies of oral health preventive programs need to consider the multifactorial etiology of childhood dental caries across LMICs to fully elucidate the impact of child–family–socioeconomic factors as drivers of dental caries.

## Figures and Tables

**Figure 1 ijerph-19-07114-f001:**
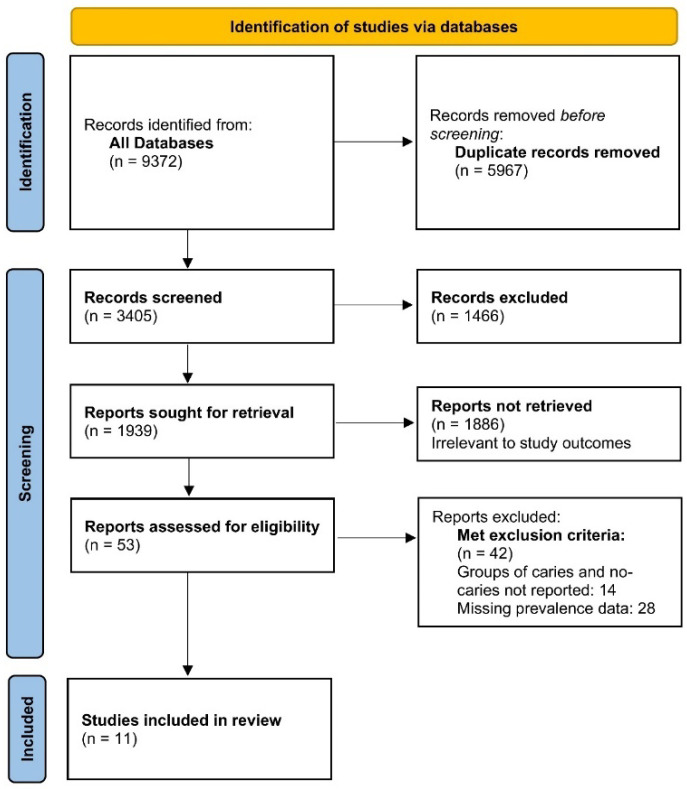
The PRISMA flow diagram.

**Figure 2 ijerph-19-07114-f002:**
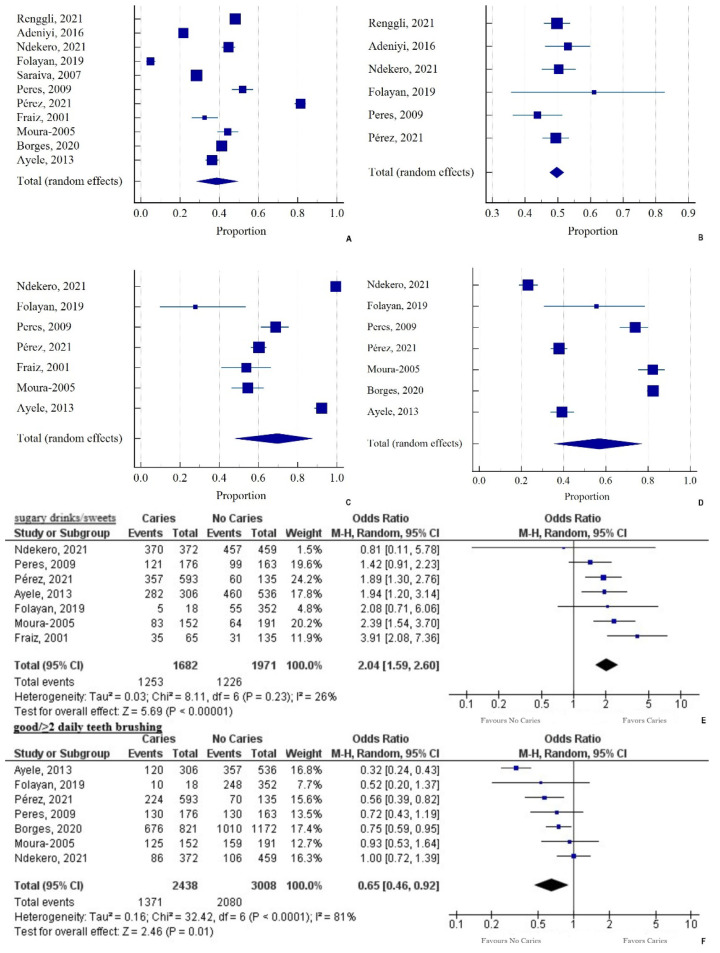
(**A**) Pooled prevalence of dental caries (N = 4250) in the total population (N = 11,115); (**B**) Pooled prevalence of females (N = 993) in the population with caries (N = 1999); (**C**) Pooled prevalence of sugary drinks/sweets consumption (N = 1253) in the population with caries (N = 1682); (**D**) Pooled prevalence of good brushing habits/≥ 2 daily teeth brushing (n = 1371/2438) in the population with caries, presented as a proportion in a forest plot, applying the random-effects model; (**E**) Sugary drinks/sweets consumption: a forest plot depicting the odds ratio (OR) of children having caries compared to no caries. Heterogeneity: Tau^2^ = 0.03; Chi^2^ = 8.11, DF = 6; I^2^ = 26%. Test for overall effect: Z = 5.69 (*p* < 0.00001); (**F**) Post good/ ≥ 2 daily teeth brushing: a forest plot depicting OR of children with caries compared to no caries. Heterogeneity: Tau^2^ = 0.16; Chi^2^ = 32.42, DF = 6 (*p* < 0.0001); I^2^ = 81%. Test for overall effect: Z = 2.46 (*p* = 0.01).

**Figure 3 ijerph-19-07114-f003:**
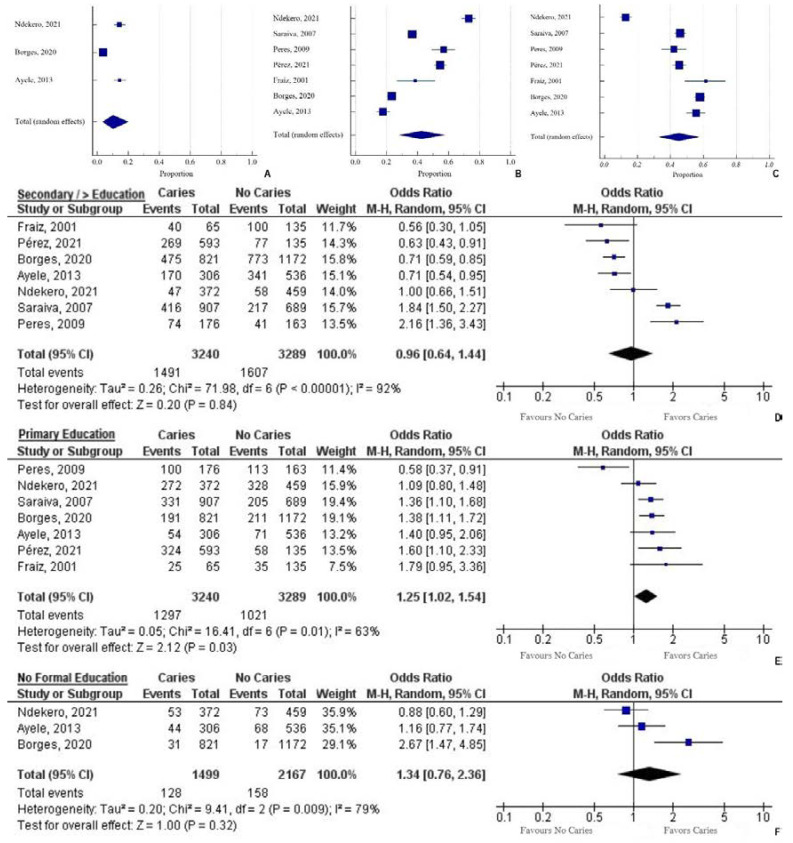
(**A**) Pooled prevalence of no formal education for mothers (n = 128/1499); (**B**) primary education for mothers (n = 1297/3240); (**C**) secondary and higher education for mothers (n = 1491/3240) of children with caries, presented as a proportion in a forest plot, applying the random-effects model. (**D**) Secondary/higher education: a forest plot depicting the odds ratio (OR) of children having caries compared to no caries. Heterogeneity: Tau^2^ = 0.26; Chi^2^ = 71.98, DF = 6; I^2^ = 92%. Test for overall effect: Z = 0.2 (*p* = 0.84); (**E**) Primary education: a forest plot depicting the odds ratio (OR) of children having caries compared to no caries. Heterogeneity: Tau^2^ = 0.05; Chi^2^ = 16.41, DF = 6; I^2^ = 63%. Test for overall effect: Z = 2.12 (*p* = 0.03); (**F**) No formal education: a forest plot depicting the odds ratio (OR) of children having caries compared to no caries. Heterogeneity: Tau^2^ = 0.2; Chi^2^ = 9.41, DF = 2; I^2^ = 79%. Test for overall effect: Z = 1 (*p* = 0.32).

**Figure 4 ijerph-19-07114-f004:**
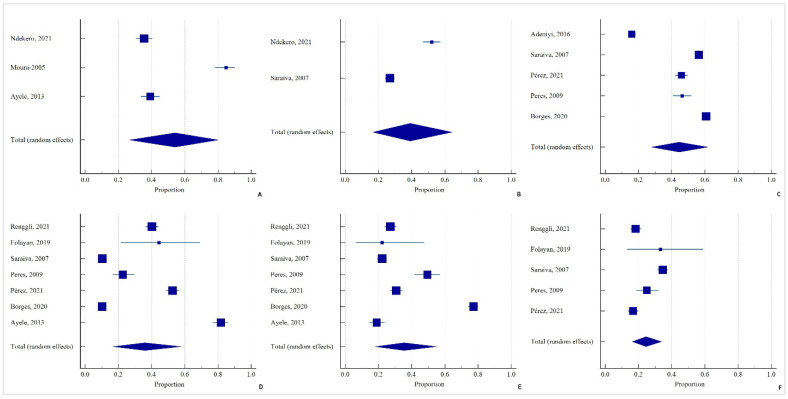
(**A**) Pooled prevalence of caregivers (i.e., mother) helping children with tooth brushing (n = 381/830) in the population with caries; (**B**) Pooled prevalence of children with caries using fluoride toothpaste (n = 438/1279); (**C**) Pooled prevalence of accessed and visited dental services in both groups (n= 2758/5629); (**D**–**F**) Pooled prevalence of low (n = 1040/3450), middle (n = 1338/3450), and high (n = 578/3450) socioeconomic status of the children/their families in the caries group, presented in a forest plot, applying the random-effects model.

**Figure 5 ijerph-19-07114-f005:**
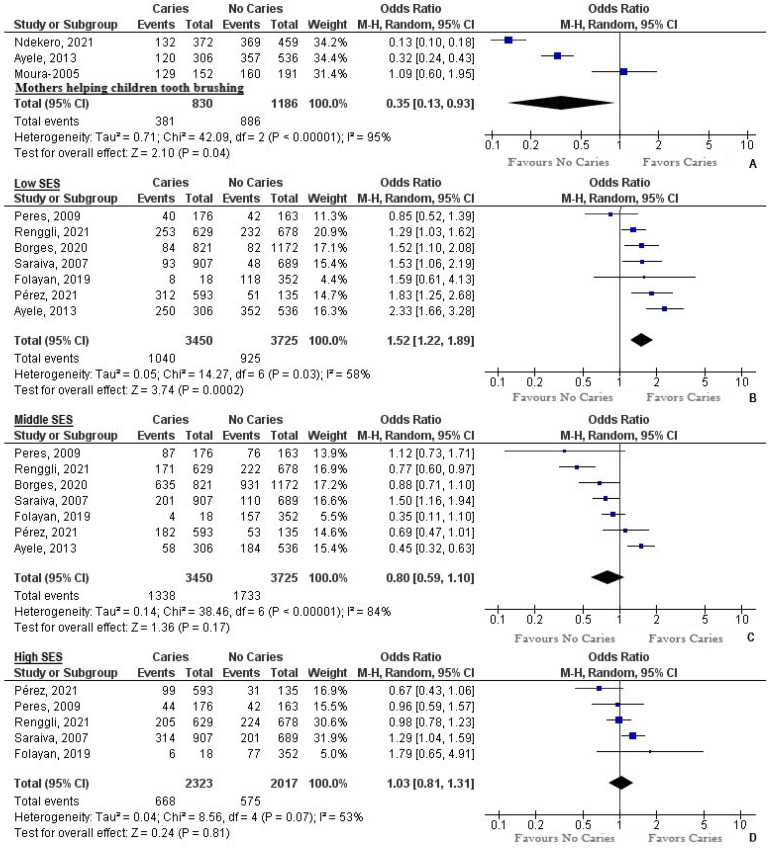
(**A**) Mothers helping children in tooth brushing: a forest plot depicting OR of children with caries compared to no caries. Heterogeneity: Tau^2^ = 0.71; Chi^2^ = 42.09, df = 2 (*p* < 0.00001); I^2^ = 95%. Test for overall effect: Z = 2.10 (*p* = 0.04); A forest plot depicting OR of children with caries compared to no caries belonging to the following socioeconomic status: (**B**) Low socioeconomic status: Heterogeneity: Tau^2^ = 0.05; Chi^2^ = 14.27, df = 6 (*p* = 0.03); I^2^ = 58%. Test for overall effect: Z = 3.74 (*p* = 0.0002) (**C**) Middle socioeconomic status: Heterogeneity: Tau^2^ = 0.14; Chi^2^ = 38.46, df = 6 (*p* < 0.00001); I^2^ = 84%. Test for overall effect: Z = 1.36 (*p* = 0.17); (**D**) High socioeconomic status: Heterogeneity: Tau^2^ = 0.04; Chi^2^ = 8.56, df = 4 (*p* = 0.07); I^2^ = 53%. Test for overall effect: Z = 0.24 (*p* = 0.81).

**Table 1 ijerph-19-07114-t001:** Baseline characteristics of the studies.

Author-Year	Type of Study	Age (Years) at Enrollment	Duration of Follow-Up (Years)	Country	Sample Size (N)	Type of Dentition	Measurements Taken	Any Dental Caries (n, %)	DMFT Scores (Mean, SD)	Anthropometric Status (n, %)	Primary Objective	NOS Score
Pérez-2021 [47]	Cross-sectional	9.9 years (SD: 1.2)	None	Mexico	728	Mixed	ICDAS II 1–6 index	Overall: 593 (81.5%); ICDAS II 1–3: 277 (38.1%); ICDAS II 4–6: 316 (43.4%)	NR	NR	To evaluate the association between sociodemographic factors and caries	7
Renggli−2021 [45]	Cross-sectional data of a prospective cohort	<2 years	1 year	Cambodia	1307	Primary	dmft index; Anthropometric measures used the WHO Child Growth Standards; Socioeconomic status using Principal Component Analysis; Dietary intake with a 24-h food recall	Overall: 629 (51.9%); <6 months: 15 (10.5%); 6–12 months: 71 (25.5%); 12–18 months: 246 (56.2%); 18–24 months: 297 (66.3%)	Overall: 5.1 (SD: 3.6); Stunted: 2.3 (SD: 3.6); Non-stunted: 5.1 (SD: 3.6) (*p* = 0.053)	Stunted: 332 (25.4%)	To examine the association between dental caries and the presence of new cases of stunting malnutrition at 1-year follow-up	8
Ndekero-2021 [46]	Cross-sectional	4.2 years (SD: 0.7)	None	Tanzania	831	Primary	dmft index; Anthropometric measures using WHO child growth standards	Overall: 372 (44.8%); 1–4 decays: 219 (26.4%); >5 decays: 101 (12.2%)	Overall: 2.5	Stunted: 13 (1.6%); Underweight: 35 (4.2%); Wasted: 248 (29.8%)	To determine the prevalence of dental caries, risk factors and nutritional status	6
Folayan-2019 [43]	Cross-sectional	3.7 years (SD: 1.3)	None	Nigeria	370	Primary	dmft index; Anthropometric measures using WHO child growth standards; OHI-S; Oral hygiene status (the index of Greene and Vermillion)	Overall: 18 (4.9%); 6–35 months: 0; 36–47 months: 5 (1.4%); 48–59 months: 8 (2.2%); 60–71 months: 5 (1.4%)	Overall: 0.14 (SD: 0.8)	Stunted: 120 (4.9%); Underweight: 20 (5.4%); Wasted: 67 (18.1%); Overweight: 20 (5.4%)	To determine the association between the prevalence of dental caries and malnutrition	7
Adeniyi-2016 [44]	Cross-sectional	7.8 years (SD: 1.5)	None	Nigeria	973	Mixed	dmft and DMFT index; Anthropometric measures using WHO child growth standards; OHI	Overall: 211 (21.7%); 5 years: 55 (5.7%); 6 years: 162 (16.6%); 7 years: 165 (17%); 8 years: 198 (20.3%); 9 years: 177 (18.2%); 10 years: 216 (22.2%)	Overall: 0.5 (SD: 1.1); 5 years: 0.2 (SD: 0.7); 6 years: 0.5 (SD: 1.3); 7 years: 0.5 (SD: 1.2); 8 years: 0.6 (SD: 1.2); 9 years: 0.6 (SD: 1.1); 10 years: 0.3 (SD: 0.8)	Stunted: 135 (13.9%); Underweight: 132 (13.6%); Wasted: 106 (10.9%)	To determine the association between caries and the nutritional status	6
Ayele-2013 [48]	Cross-sectional	7–14 years	None	Ethiopia	842	Mixed	dmft and DMFT index	Overall: 306 (36.3%)	NR	NR	To assess the prevalence and associated factors of dental caries	5
Borges-2012 [42]	Cross-sectional	4–6 years	None	Brazil	1993	Primary	dmft index	Overall: 821 (41.2%)	Overall: 1.5 (SD: 2.6)	NR	To analyze the influence of socio-behavioral factors on the prevalence and severity of dental caries	5
Saraiva-2007 [39]	Cross-sectional	2–5 years	None	Brazil	3189	Primary	dmft index; Anthropometric measures using WHO child growth standards	Overall: 907 (28.4%); >1: 689 (21.6%)	NR	NR	To assess the association between intrauterine growth restriction and dental caries	5
Moura-2006 [40]	Cross-sectional data of a prospective cohort	3–6 years	0–3 years	Brazil	343	Primary	dmft index	Overall: 152 (44.3%)	2.1 (SD: 1.4)	NR	To evaluate the prevalence of caries in children that participate in a dental program attending mothers and children	4
Peres-2005 [38]	Prospective cohort	Birth	12 years	Brazil	339	Mixed	dmft and DMFT index; Anthropometric measures using WHO child growth standards	Overall: 176 (51.8%)	Overall: 1.2 (SD: 1.6)	Height for age (HAZ) at 1 year >2 (caries vs. no caries): 149 (84.7%) vs. 154 (94.5%); ≤2 (caries vs. no caries): 23 (13.1%) vs. 5 (3.1%); Height for age (HAZ) at 4 years >2 (caries vs. no caries): 149 (84.7%) vs. 155 (95.1%); ≤2 (caries vs. no caries): 23 (13.1%) vs. 5 (3.1%)	To investigate the relationship between social and biological conditions experienced in very early life	7
Fraiz-2001 [41]	Cross-sectional data of a prospective cohort	2.9 years (SD: 0.6)	1 year	Brazil	200	Primary	dmft index	65 (32.5%)	NR	NR	To investigate the factors associated with the development of dental caries in preschool children who receive regular dental care and follow-up	6

**Table 2 ijerph-19-07114-t002:** Child-level characteristics of eleven studies.

Author-year	Gender (n, %)		Ever Breastfed (n, %)		Ever Bottle-fed (n, %)		Bottle-Fed at Night (n, %)		Nutritional Patterns (n, %)		Brushing Patterns/Oral Hygiene Status (n, %)	
Pérez-2021 [47]	**Caries:** 593	**No caries:** 135	**Caries:** 593	**No caries:** 135	**Caries:** 593	**No caries:** 135	**Caries:** 593	**No caries:** 135	**Caries:** 593	**No caries:** 135	**Caries:** 593	**No caries:** 135
	Male: 300 (50.6%); Female: 293 (49.4%)	Male: 71 (52.6%); Female: 64 (47.4%)	NR		NR		NR		Sweets consumption: > Once per day: 357 (60.2%); Seldom/sometimes per week: 236 (39.8%); Soft drinks consumption: > Once per day: 348 (58.7%); Seldom/sometimes per week: 245 (41.3%)	Sweets consumption: > Once per day: 60 (44.4%); Seldom/sometimes per week: 75 (55.6%); Soft drinks consumption: > Once per day: 75 (55.6%); Seldom/sometimes per week: 60 (44.4%)	Brushing frequency: Less than twice daily: 369 (62.2%); Twice or more daily: 224 (37.8%); Oral hygiene: Good: 279 (47%); Poor: 314 (53%)	Brushing frequency: Less than twice daily: 65 (48.2%); Twice or more daily: 70 (51.8%); Oral hygiene: Good: 81 (60%); Poor: 54 (40%)
Regnnli-2021 [45]	**Caries:** 629	**No caries:** 678	NR		NR		NR		Minimum acceptable diet: 640 (49.0%)		NR	
	Male: 316 (50.2%); Female: 313 (49.9%)	Male: 315 (46.5%); Female: 363 (53.5%)										
Ndekero-2021 [46]	**Caries:** 372	**No caries:** 459	NR		NR		NR		**Caries:** 372	**No caries:** 459	**Caries:** 372	**No caries:** 459
	Male: 185 (49.7%); Female: 187 (50.3%)	Male: 214 (46.6%); Female: 245 (53.4%)							Sugary foods in between meals: 346 (93%); Drink juice or sodas or sugary drinks: 370 (99.5%); Eat fruits: 260 (69.9%) vs. 312 (68%); Eat vegetables: 369 (99.1%)	Sugary foods in between meals: 423 (92.2%); Drink juice or sodas or sugary drinks: 457 (99.6%); Eat fruits: 312 (68%); Eat vegetables: 450 (98%)	Not every day: 11 (3%); Once a day: 275 (73.9%); Twice or more daily: 86 (23.1%)	Not every day: 13 (2.8%); Once a day: 337 (73.4%); Twice or more daily: 106 (23.1%)
Folayan-2019 [43]	**Caries:** 18	**No caries:** 352	NR		NR		NR		**Caries:** 18	**No caries:** 352	**Caries:** 18	**No caries:** 352
	Male: 7 (38.9%); Female: 11 (61.1%)	Male: 196 (55.7%); Female: 156 (44.3%)							≥3 times daily sugar consumption between meals: 5 (27.7%); <3 times daily sugar consumption between meals: 13 (72.2%)	≥3 times daily sugar consumption between meals: 55 (15.6%); <3 times daily sugar consumption between meals: 297 (84.4%)	Poor: 0; Fair: 8 (44.4%); Good: 10 (65.6%)	Poor: 8 (2.3%); Fair: 96 (27.3%); Good: 248 (70.5%)
Adeniyi-2016 [44]	**Caries:** 211	**No caries:** 762	NR		NR		NR		NR		NR	
	Male: 99 (49.9%); Female: 112 (53.1%)	Male: 389 (51%); Female: 373 (49%)										
Ayele-2013 [48]	Male: 379 (45%); Female: 463 (55%)		NR		NR		NR		**Caries:** 306	**No caries:** 356	**Caries:** 306	**No caries:** 356
									Snack frequency: Thrice/day: 178 (58.1%); Twice/day: 11 (3.6%); Once/day: 81 (26.5%); Occasional: 36 (11.8%); Sweet foods and drinks: 282 (92.2%); Soft drinks: 224 (73.2%); Sugared coffee: 175 (57.2%)	Snack frequency: Thrice/day: 346 (64.5%); Twice/day: 23 (4.3%); Once/day: 116 (21.6%); Occasional: 51 (9.5%); Sweet foods and drinks: 460 (85.8%); Soft drinks: 60 (11.2%); Sugared coffee: 303 (56.5%)	Rinsing mouth: 260 (85%); Cleaning teeth: 120 (39.2%)	Rinsing mouth: 49 (9.1%); Cleaning teeth: 357 (66.6%)
Borges-2012 [42]	Male: 984 (49.4%)		NR		NR		NR		NR		**Caries:** 821	**No caries:** 1172
											Not at all or once a day: 91 (11.1%); Twice or more daily: 676 (82.3%)	Not at all or once a day: 108 (9.2%); Twice or more daily: 1010 (86.2%)
Saraiva-2007 [39]	NR		**Caries:** 907	**No caries:** 689	**Caries:** 907	**No caries:** 689	NR		**Caries:** 907	**No caries:** 689	NR	
			Yes: 158 (17.4%)	Yes: 133 (19.3%)	≤19 months: 177 (19.5%); >19 months: 230 (25.4%)	≤19 months: 96 (14%); >19 months: 134 (19.4%)			Carbohydrate intake: <161 g/day: 212 (23.4%); 161–249.1 g/day: 426 (47%); >249.2 g/day: 259 (28.5%); Sucrose intake: <35 g/day: 178 (19.6%); ≥35 g/day: 267 (29.4%)	Carbohydrate intake: <161 g/day: 130 (18.8%); 161–249.1 g/day: 242 (35.1%); >249.2 g/day; 135 (19.6%); Sucrose intake: <35 g/day: 76 (11%); ≥35 g/day: 90 (13.1%)		
Moura-2006 [40]	Male: 169 (49.2%)		NR		NR		NR		**Caries:** 152	**No caries:** 191	**Caries:** 152	**No caries:** 191
									Daily sugar consumption: Always: 83 (54.6%); Sometimes: 64 (42.1%); Never: 4 (2.6%)	Daily sugar consumption: Always: 64 (33.5%); Sometimes: 16 (8.4%); Never: 11 (5.8%)	1/day: 27 (17.8%); 2/day: 70 (46.1%); ≥3/day: 55 (36.2%); Brushing before going to sleep: Yes: 103 (67.8%); No: 48 (31.6%)	1/day: 32 (16.8%); 2/day: 82 (42.9%); ≥3/day: 77 (40.3%); Brushing before going to sleep: Yes: 132 (69.1%); No: 59 (30.9%)
Peres- 2005 [38]	**Caries:** 176		**No caries:** 163		NR	NR	NR	**Caries:** 176	**No caries:** 163	**Caries:** 176	**No caries:** 163	
	Male: 98 (55.7%); Female: 77 (43.8%)		Male: 84 (51.5%); Female: 80 (49.1%)					Sweet consumption: Almost never/less than once a day: 54 (30.7%); At least once daily: 121 (68.8%)	Sweet consumption: Almost never/less than once a day: 60 (36.8%); At least once daily: 99 (60.7%)	Brushing frequency: ≥2: 130 (73.9%); <2: 45 (25.6%); Child brushed teeth at emergence of 1st teeth: 39 (2; 2.2%); Child brushed teeth after 1 year: 100 (56.8%)	Brushing frequency: ≥2: 130 (79.8%); <2: 34 (20.9%); Child brushed teeth at emergence of 1st teeth: 59 (36.2%); Child brushed teeth after 1 year: 79 (48.5%)	
Fraiz-2001 [41]	NR		NR		**Caries:** 65	**No caries:** 135	**Caries:** 65	**No caries:** 135	**Caries:** 65	**No caries:** 135	NR	
					Bottle-fed: 55 (84.6%); Never bottle-fed: 4 (6.2%); No longer bottle-fed: 6 (9.2%)	Bottle-fed: 107 (79.3%); Never bottled-fed: 13 (9.6%); No longer bottle-fed: 15 (11.1%)	Never to sleep: 22 (33.8%); To sleep: 23 (35.4%); Sleeping: 10 (15.4%) Not bottle-fed: 10 (15.4%)	Never to sleep: 78 (57.8%); To sleep: 23 (17%); Sleeping: 6 (4.4%); Not bottle-fed: 28 (20.7%)	Sugar consumption: High: 35 (53.8%); Moderate: 30 (46.2%)	Sugar consumption: High: 31 (23%); Moderate: 104 (77%)		

**Table 3 ijerph-19-07114-t003:** Family-level characteristics of eight studies.

Author-Year	Maternal Age (Years)		Parental Educational Status (n, %)		Help with Child Tooth Brushing (n, %)		Fluoride Toothpaste (n, %)	
Pérez-2021 [47]	NR		**Caries:** 593	**No caries:** 135	NR		NR	
			≥9 years: 269 (45.4%); <9 years: 324 (54.6%)	≥9 years: 77 (57%); <9 years: 58 (43%)				
Ndekero-2021 [46]	NR		**Caries:** 372	**No caries:** 459	**Caries:** 372	**No caries:** 459	**Caries:** 352	**No caries:** 459
			Maternal: Informal and primary education: 272 (73.1%); Secondary education and above: 47 (12.6%); No education: 53 (14.8%)	Maternal: Informal and primary education: 328 (71.5%); Secondary education and above: 58 (12.6%); No education: 73 (15.9%)	Yes: 132 (35.5%)	Yes: 369 (80.4%)	194 (52.2%)	235 (51.2%)
Ayele-2013 [48]	NR		**Caries:** 306	**No caries:** 356	**Caries:** 306	**No caries:** 356	NR	
			Paternal: Illiterate: 44 (14.4%); Read & write: 38 (12.4%); 1–6 grade: 54 (17.6%); 7–12 grade: 95 (31%); >12th grade: 75 (24.5%)	Paternal: Illiterate: 68 (12.7%); Read & write: 56 (10.4%); 1–6 grade: 71 (13.2%); 7–12 grade: 148 (27.6%); >12th grade: 193 (36%)	Yes: 120 (39.2%); No: 186 (60.8%)	Yes: 357 (66.6%); No: 179 (33.4%)		
Borges-2012 [42]	NR		**Caries:** 821	**No caries:** 1172	NR		NR	
			Parents: Illiterate: 31 (3.8%); Elementary school: 191 (23.3%); High school: 413 (50.3%) University: 62 (7.6%)	Parents: Illiterate: 17 (1.5%); Elementary school: 211 (18%); High school: 647 (55.2%); University: 126 (10.7%)				
Saraiva-2007 [39]	**Caries:** 907	**No caries:** 689	**Caries:** 907	**No caries:** 689	NR		**Caries:** 907	**No caries:** 689
	<20 years: 138 (15.2%); 20–29 years: 201 (22.2%); >29 years: 141 (15.5%)	<20 years: 147 (21.4%); 20–29 years: 114 (16.6%); >29 years: 74 (10.7%)	Maternal: >12 years of education: 181 (19.9%); 12 years: 235 (25.9%); <12 years: 331 (36.5%)	Maternal: >12 years of education: 85 (12.4%); 12 years: 132 (19.2%); <12 years: 205 (29.7%)			244 (26.9%)	132 (19.1%)
Moura-2006 [40]	NR		NR		**Caries:** 152	**No caries:** 191	NR	
					Yes: 129 (84.9%)	Yes: 160 (83.8%)		
Peres- 2005 [38]	NR		**Caries:** 176	**No caries:** 163	NR		NR	
			Maternal: ≥8 years of education: 74 (42.3%); <8 years of education: 100 (56.6%); Paternal: ≥8 years of education: 33 (18.8%); <8 years of education: 147 (83.5%)	Maternal: ≥8 years of education: 41 (25.2%); <8 years of education: 113 (69.3%); Paternal: ≥8 years of education: 45 (27.6%); <8 years of education: 79 (48.5%)				
Fraiz-2001 [41]	26.3 years (SD: 5.1)		**Caries:** 65	**No caries:** 135	NR		NR	
			Maternal: ≤8: 25 (38.5%); >8: 40 (61.5%); Paternal: ≤8: 28 (43.1%); >8: 32 (49.2%)	Maternal: ≤8: 35 (25.9%); >8: 100 (74.1%); Paternal: ≤8: 37 (27.4%); >8: 87 (64.4%)				

**Table 4 ijerph-19-07114-t004:** Socioeconomic-level characteristics of eleven studies.

Author-Year	SES (n, %)		Setting (n, %)		Access to/Visited Dental Services (n, %)		Additional Comments	
Pérez-2021 [47]	**Caries:** 593	**No caries:** 135	Public schools		**Caries:** 593	**No caries:** 135	-	
	SES: Low: 312 (52.6%); Middle: 182 (30.7%); High: 99 (16.7%)	SES: Low: 51 (37.8%); Middle: 53 (39.2%); High: 31 (23%)			Yes: 260 (43.8%)	Yes: 74 (54.8%)		
Regnnli-2021 [45]	**Caries:** 629	**No caries:** 678	**Caries:** 629	**No caries:** 678	NR		-	
	SES: Lowest: 118 (9%); Low: 135 (10.3); Medium: 171 (13.1%); High: 115 (8.8%); Highest: 90 (6.8%)	SES: Lowest: 102 (7.8%); Low: 130 (9.9%); Medium: 222 (16.9%); High: 100 (7.6%); Highest: 124 (9.5%)	Rural: 476 (36.4); Urban: 153 (11.7)	Rural: 512 (39.2%); Urban: 166 (12.7%)				
Ndekero-2021 [46]	NR		**Caries:** 372	**No caries:** 459	NR		**Caries:** 372	**No caries:** 459
			Rural: 315 (84.7%) vs. 346 (75.4%); Semi-rural: 57 (15.3%) vs. 113 (24.6%)	Rural: 346 (75.4%); Semi-rural: 113 (24.6%)			Siblings present: 321 (86.3%); Mother’s not formally employed: 358 (96.2%); Difficulty in purchasing food for child due to costs: 122 (32.8%)	Siblings present: 403 (87.8%); Mother’s not formally employed: 35 (7.6%); Difficulty in purchasing food for the child due to costs: 156 (34%)
Folayan-2019 [43]	**Caries:** 18	**No caries:** 352	Peri-urban households		NR		Mean oral hygiene score: 1.1 (SD: 1.2) ~ good	
	Low: 8 (44.4%); Middle: 4 (22.2%); High: 6 (33.3%)	Low: 118 (33.5%); Middle: 157 (44.6%); High: 77 (21.9%)						
Adeniyi-2016 [44]	NR		**Caries:** 211	**No caries:** 762	Yes: 155 (15.9%); No: 818 (84.1%)		OHI: 0.4 (SD: 1.1) ~ good: 608 (62.5%); 0.6 (SD = 1.2) ~ fair: 365 (37.5%)	
			Public school: 56 (26.5%); Private school: 55 (26.1%)	Public school: 505 (66.3%); Private school: 257 (33.7%)				
Ayele-2013 [48]	**Caries:** 306	**No caries:** 356	Rural and urban households		NR		-	
	<28 USD: 157 (51.3%); 29–56 USD: 93 (30.4%); 57–84 USD: 20 (6.5%); 85–112 USD: 24 (7.8%); 113–167 USD: 8 (2.6%); >168 USD: 4 (1.3%)	<28 USD: 206 (38.4%); 29–56 USD: 146 (27.2%); 57–84 USD: 46 (8.6%); 85–112 USD: 69 (12.9%); 113–167 USD: 38 (7.1%); >168 USD: 31 (5.8%)						
Borges-2012 [42]	**Caries:** 821	**No caries:** 1172	Public preschools		**Caries:** 821	**No caries:** 1172	-	
	<1 minimum wage: 84 (10.2%); 1–1.9 minimum wage: 366 (44.6%); 2–2.9 minimum wage: 163 (19.9%); ≥ 3 minimum wage: 106 (12.9%)	<1 minimum wage: 82 (7%); 1–1.9 minimum wage: 456 (38.9%); 2–2.9 minimum wage: 241 (20.6%); ≥ 3 minimum wage: 234 (20%)			Yes: 552 (67.2%); No: 265 (32.3%)	Yes: 659 (56.2%); No: 512 (43.7%)		
Saraiva-2007 [39]	**Caries:** 907	**No caries:** 689	NR		**Caries:** 907	**No caries:** 689	**Caries:** 907	**No caries:** 689
	Poverty ratio: >3.5: 93 (10.3%); 1.301–3.5: 201 (22.2%); <1.301: 314 (34.6%)	Poverty ratio: >3.5: 48 (7.7%); 1.3–3.5: 110 (16%); <1.3: 201 (29.2%)			Yes: 570 (62.8%)	331 (48.1%)	Passive smoking: 483 (53.2%)	Passive smoking: 280 (40.7%)
Moura-2006 [40]	NR		Clinic		**Caries:** 152	**No caries:** 191	Data were obtained at follow-up after participation in the Preventive Program for Pregnant Mothers and Babies whose goals are to recover and maintain oral health in pregnant women and children aged 0–3 years	
					1–6 months ago: 65 (42.8%); 6–12 months ago: 29 (19.1%); Over 12 months ago: 57 (37.5%)	1–6 months ago: 62 (32.5%); 6–12 months ago: 44 (23%); Over 12 months ago: 85 (44.5%)		
Peres- 2005 [38]	**Caries:** 176	**No caries:** 163			**Caries:** 176	**No caries:** 163	The same participants were followed at 6 and 12 years of age and caries are reported for the second follow-up at age 12; Piped water supply: Yes: 143 (81.3%) vs. 141 (86.5%); No: 30 (17%) vs. 21 (12.9%); Adequate birth weight and gestational age: Yes: 30 (17%) vs. 20 (12.3%); No: 113 (64.2%) vs. 116 (71.7%)	
	Income: 1st quartile: 40 (22.7%); 2nd quartile: 41 (23.3%); 3rd quartile: 46 (26.1%); 4th quartile: 44 (25%); Social class: Employers/Professional: 30 (17%); Skilled workers: 119 (67.6%); Unskilled workers: 11 (6.3%)	Income: 1st quartile: 42 (25.8%); 2nd quartile: 40 (24.5%); 3rd quartile: 36 (22.1%); 4th quartile: 42 (25.8%); Social class: Employers/Professional: 41 (25.2%); Skilled workers: 99 (60.7%); Unskilled workers: 6 (3.7%)	Urban households		Yes: 85 (48.3%); No: 90 (51.1%)	Yes: 72 (44.2%); No: 92 (56.4%)		
Fraiz-2001 [41]	NR		Clinic		NR		Children aged 1–2 years and mothers, who had already taken part in a dental program at a clinic during, at least, the previous twelve months were enrolled	

## Data Availability

All data was sourced from public repositories.

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
