# Peer review of "Individual, Family, and Socioeconomic Contributors to Dental Caries in Children from Low- and Middle-Income Countries"

_ijerph, 2022, doi:10.3390/ijerph19127114_

Round 1

Reviewer 1 Report

This is a systematic review, the subject is original and very relevant. The methodology was very well outlined and described, PRISMA checklist can be verified in the text . Inclusion and exclusion criteria are clear.  the results are very well presented , the discussion and conclusion are justified by the  results.

One  suggestion to the authors is to decrease the number of tables or figures.

Author Response

REVIEWER 1 COMMENTS AND RESPONSES

Reviewer 1

Comment 1:

This is a systematic review; the subject is original and very relevant. The methodology was very well outlined and described, PRISMA checklist can be verified in the text. Inclusion and exclusion criteria are clear.  The results are very well presented, the discussion and conclusion are justified by the results.

Response 1:

Thank you very much for the thorough feedback!                                     

Comment 2:

One suggestion to the authors is to decrease the number of tables or figures.

Response 2:

Thank you for the meaningful insight. We have reduced the total number of figures in our manuscript. All additional tables have been reported under supplementary tables.

Reviewer 2 Report

Dear Authors, 

this paper is really interesting and it is helpful to the reader to understand the situation about the topic in low and middle.income countries.

By the way, some issues need to be solved before publication.

Abstract: it is well written and it well summarizes the research that has been conducted.

Introduction: It is really to short. In a systematic review, but in papers in general, introduction should describe the overall situation about the main topic of the research. In your article, this part is really poor. i strongly recommend to add some information about the difference between caries and "dmft" situation between low, middle and high income countries; some information about how instruction could help improve oral hygiene in children in order to prevent caries some more information should be added regarding the differences between different countries in other pediatric dentistry fields, such as traumatology ( for this topic i suggest you to use this paper: Ludovichetti FS, Signoriello AG, Zuccon A, Padovani S, Mazzoleni S. The Role of Information in Dental Traumatology in Patients during Developmental Age: A Cognitive Investigation. Eur J Dent. 2021 Oct 22. Some informations  should also be added at the end of the introduction regarding the existing different methods of treating dental caries in different countries (for this topic i suggest you to use this reference: Ludovichetti FS, Stellini E, Signoriello AG, DI Fiore A, Gracco A, Mazzoleni S. Zirconia vs. stainless steel pediatric crowns: a literature review. Minerva Dent Oral Sci. 2021 Jun;70(3):112-118.

Methods: perfectly performed and well explained.

Results: perfectly performed and well explained.

Discussion: Line 309-310: please, be sure about this statement, double check it. 

Line 370-380: please, clarify better the role of fluoridated water and fluoride toothpastes in preventing cries and the differences between low and middle income counties and high-income countries. 

Line 385-395: please, describe better your findings and thoughts about the impact that maternal age and maternal instruction could have in the caries presence in children.

Line 396: this chapter is really to long, please shorten it.

Author Response

REVIEWER 2 COMMENTS AND RESPONSES

Reviewer 2

Comment 1:

Dear Authors, this paper is really interesting and it is helpful to the reader to understand the situation about the topic in low and middle income countries.

Response 1:

Thank you for taking out time to provide a detailed helpful feedback!

Comment 2: By the way, some issues need to be solved before publication. Abstract: it is well written and it well summarizes the research that has been conducted.

Response 2: Noted, thank you!

Comment 3:

Introduction: It is really too short. In a systematic review, but in papers in general, introduction should describe the overall situation about the main topic of the research. In your article, this part is really poor.

1) I strongly recommend to add some information about the difference between caries and "dmft" situation between low, middle and high income countries.

2) Some information about how instruction could help improve oral hygiene in children in order to prevent caries.

3) Some more information should be added regarding the differences between different countries in other pediatric dentistry fields, such as traumatology (for this topic i suggest you to use this paper: Ludovichetti FS, Signoriello AG, Zuccon A, Padovani S, Mazzoleni S. The Role of Information in Dental Traumatology in Patients during Developmental Age: A Cognitive Investigation. Eur J Dent. 2021 Oct 22.

4) Some information should also be added at the end of the introduction regarding the existing different methods of treating dental caries in different countries (for this topic i suggest you to use this reference: Ludovichetti FS, Stellini E, Signoriello AG, DI Fiore A, Gracco A, Mazzoleni S. Zirconia vs. stainless steel pediatric crowns: a literature review. Minerva Dent Oral Sci. 2021 Jun;70(3):112-118.

Response 3:

  • Differences between caries/dmft between low-, middle-, and high-income countries have been summarized in lines 50-53:

The World Health Organization (WHO) data suggests that the decline in the prevalence of caries in the first decade of this century is only noted in HICs e.g. USA and Western Europe, whereas LMICs have either had less explicit reduction or increase in burden in certain countries e.g. Gambia, Croatia, Saudi Arabia, and Moldova. 4

  • Information of improvement of oral hygiene has also been added in lines 64-67:

It is necessary to establish important prophylactic measure such as developing regular oral hygiene e.g. proper brushing and flossing methods, fluoride application, minimizing sugar intake, and improvement of health education among primary caregivers.17,18

  • Differences between different countries in pediatric dentistry fields have been summarized and the article by Ludovichetti et al. (2021) has been also been cited in lines 54-57:

Regional differences in oral health disparities for children are also noted in quality of care, access to care, cost of care, oral health education, and health literacy.5–7 Majority of the health service expenditures in HICs is invested in preventive oral care whereas it is used for emergency oral care and pain relief in LMICs.8

  • Differences in dental practices for treatment of dental caries have been noted as well as adding the recommended article by Ludovichetti et al. (2021) in lines 126-131:

Dental caries requires rigorous assessment of caries risk and carious lesion activity for a focused treatment plan which may vary depending on different dental practices.29 For instance, zirconia crowns (ZC) are being used due to their acceptable efficacy and aesthetic acceptance compared to the traditionally used stainless steel crowns (SSC) for primary teeth restoration.30 Carious teeth treatment is evolving, albeit predominantly in HICs, and is necessary to improve dental care in children.31

Comment 4:

Methods: perfectly performed and well explained. Results: perfectly performed and well explained.

Response 4:

Noted, thanks!

Comment 5:

  • Discussion: Line 309-310: please, be sure about this statement, double check it.
  • Line 370-380: please, clarify better the role of fluoridated water and fluoride toothpastes in preventing cries and the differences between low and middle income counties and high-income countries.
  • Line 385-395: please, describe better your findings and thoughts about the impact that maternal age and maternal instruction could have in the caries presence in children.
  • Line 396: this chapter is really too long, please shorten it.

Response 5:

  • This is not the first review and the statement has been changed in lines 322-323:

“We conducted a systematic review and meta-analysis to explore the prevalence and associated factors of dental caries in children residing in LMICs.”

  • The role of fluoridated water/toothpaste in preventing caries have been summarized in lines 385-388:

“Fluoride can improve dental health topically e.g. toothpaste, mouth rinse, topical treatments in dental clinics, and systemically e.g. water fluoridation, salt, fluoride supplementation; both types of applications are known to improve oral health yet topical application is the most effective.69

Differences in different countries have been summarized in lines 392-396:

However, a study from Brazil highlighted the discrepancy of fluoridated tap water only in better-off towns within the country.73 Countries that have not yet implemented fluoridated programs are primarily from LMICs and require technical assistance and guidance to execute population-wide automated measures.74

  • Impact of maternal education and maternal age has been added in lines 408-416:

“Maternal education ties in closely with attitude, perception, and family environmental influence on children’s oral health practices.79 For instance, parents with poor oral health are more likely to have children with dental caries, perhaps due to the interaction between genetic and environmental exposures.80 Maternal age has been found to have a u-shaped relationship e.g. mothers under 25 years or over 34 years of age at the birth of child, with dental caries among children.81 The mechanism between maternal age and effect on dental caries is likely due to different underlying factors among older and younger mothers – social factors, baseline health, and health behaviors.81

  • A few lines have been removed to shorten the length.

Reviewer 3 Report

The topic is interesting but the methodology and presentation need to be improved

Fig 1 showed 12 papers included in the study but Table 1 and 2 demonstrated only 11 papers baseline and 'infant' charateristics, where was the other paper

The title and many places in the manuscript mentioned 'infant' but the age group of the reviewed papers was up to 12 years old children, not infant any more, only the paper from Cambodia was <2 years old as infant.

The caries experience used DMFT was confused whether caries in permanent teeth only since for primary teeth it should be dmft rather.

In Table 1, type of dentition showed mixed dentition but caries measurement as DMFT which was not relevant since mixed dentition should demonstrate both primary (dmft) and permanent  (DMFT) caries measurement. Moreover, the paper from Mexico (Ref 37) used ICDAS but the paper did not mention which ICDAS 1-3 or 4-6 or 1-6 as cariesin Table 2.

Table 1. data from Nigeria (ref 33) showed exceptional low caries (dmft 014 at 3-7 years old children) which was out of limited with very low number of caries children (only 18 children) therrefore inclusion of this data in the meta analysis was questionable.

Suggestion that the papers included should be only for primary teeth caries which were the majority of the papers included and inline witht eh title of the paper meentioed 'infant' which in reality should be 'pre-school children' rather. 

Fig 3 the forest plot showed sugary drinks favoured no caries whereas good daily tooth-brushing favoured caries which was not logic, similar to Fig 5 conclusion. I may be wrong???

One of the limitation of this paper was that LMIC only covered 7 countries and therefore the result could not represent all LMICs children. 

There were some confusion on 'infant caries', 'early childhood caries' which seemed as the main interest of this paper but the reviewed and included papers (11 or 12 papers) covered up to 12 years which was not early childhood or infant at all.

Author Response

REVIEWER 3 COMMENTS AND RESPONSES

Reviewer 3

Comment 1:

The topic is interesting but the methodology and presentation need to be improved. Fig 1 showed 12 papers included in the study but Table 1 and 2 demonstrated only 11 papers, where was the other paper.

Response 1:

Thank you for pointing out the error. It is a typographical error when making the Figure 1. This has been updated and the manuscript reflects the same.

Comment 2:

The title and many places in the manuscript mentioned 'infant' but the age group of the reviewed papers was up to 12 years old children, not infant any more, only the paper from Cambodia was <2 years old as infant.

Response 2:

Thank you for your valuable insight. The correct term is “children” or “child” and it has been adjusted to reflect as such in our manuscript. The title has been changed to “individual”.

Comment 3:

The caries experience used DMFT was confused whether caries in permanent teeth only since for primary teeth it should be dmft rather.

Response 3:

Both indices were used for either primary or permanent dentition in children. Necessary changes are made on lines 99-102:

“The dmft (decayed, missing, and filled primary teeth) and DMFT (decayed, missing, and filled permanent teeth) index was used as a global measure for the assessment of oral health via categorization as the number of decayed teeth, the number of treated teeth, and the number of teeth missing due to decay.34

Comment 4: In Table 1, type of dentition showed mixed dentition but caries measurement as DMFT which was not relevant since mixed dentition should demonstrate both primary (dmft) and permanent (DMFT) caries measurement. Moreover, the paper from Mexico (Ref 37) used ICDAS but the paper did not mention which ICDAS 1-3 or 4-6 or 1-6 as caries in Table 2.

Response 4:

The dmft and DMFT index were both used as measurement for mixed dentition and ICDAS 1-6 was used by the Mexican study. All these changes are made in table 1 and 2 as well as in text to remove any confusion.

Comment 5:

Table 1. data from Nigeria (ref 33) showed exceptional low caries (dmft 014 at 3-7 years old children) which was out of limited with very low number of caries children (only 18 children) therefore inclusion of this data in the meta analysis was questionable.

Response 5:

Respected Reviewer: Thank you for pointing that out. The studies were included against the inclusion criteria “Studies were included if they contained measures of child-level, family-level, and socioeconomic-level factors contributing to dental caries in children residing in low- and middle-income countries (LMICs) as defined by the World Bank.”  While the study does have low caries in the sample population, the study is valuable as we can then understand what i) child-level, ii) family-level, and iii) socioeconomic factors led to dental caries. Such data is all the more valuable in such meta-analytical studies, as they fit our pre-identified criteria. Please review the inclusion criteria as it may help clear the confusion.

Comment 6: Suggestion that the papers included should be only for primary teeth caries which were the majority of the papers included and inline with the title of the paper mentioned 'infant' which in reality should be 'pre-school children' rather.

Response 6:

We appreciate your suggestion immensely. We did consider only primary teeth caries initially but we consider dental caries in either primary or permanent dentition as the focus of our paper in children – not only infants. These have been clarified throughout the text and the term infant  has been removed entirely in favor of children.

Comment 7:

Fig 3 the forest plot showed sugary drinks favored no caries whereas good daily tooth-brushing favored caries which was not logic, similar to Fig 5 conclusion. I may be wrong?

Reviewer 7:

Whenever an Odds Ratio plot is computed, the cutoff for the plot is “0.” In this case, the Odds Ratio value is 0.65, which favors “No Caries.” However, the label of the plot was incorrect. This has been fixed. Please review.

Comment 8: One of the limitation of this paper was that LMIC only covered 7 countries and therefore the result could not represent all LMICs children.

Response 8:

This has been acknowledged as a limitation in lines 426-429:

Another important limitation was that we obtained data from six countries (Mexico, Brazil, Cambodia, Tanzania, Nigeria, Ethiopia) across Latin America and Africa. Therefore, we do not expect our findings to be generalizable across LMICs, particularly in other regions.”

Comment 9:

There was some confusion on 'infant caries', 'early childhood caries' which seemed as the main interest of this paper but the reviewed and included papers (11 or 12 papers) covered up to 12 years which was not early childhood or infant at all.

Response 9:

This is correct – we have corrected our manuscript and made changes to reflect child-level characteristics and these terms have been removed to eliminate any confusion whatsoever.

Round 2

Reviewer 3 Report

Abstract  p 1 ln 29 secondary 'education' reduced。

Fig 1 p 4 column Records Excluded should mention what was the reason for excluded

Data extraction p 5 ln 142 dmft/DMFT score  

Study Characteristics p 5 ln 188 (N=11,115)

Table 1 p 7 Peres 2005 what is HAZ?

Fig 2 p 8 forest plot depicting the Odd ratio on sugary drink/sweet consumption 0.1-0.5 should be 'Favours no caries' whereas 1-10 should be 'Favours caries' (similar to the one on good toothbrushing on the lowest bottom graph) i.e. Favours caries on the right side and Favours no caries on the left side

Fig 3 p 13 ln 264-265 it should mention which one defined A,B, C in the upper graph series, also there was errors defining the forest plot depicting OR  i.e. Favours caries on the right side and Favours no caries on the left side (of all 3 lower graphs)

Fig 5 p 15 also similar errors i.e. Favours caries on the right side and  Favours no caries on the lest side (of all 4 graphs)

Linitations and recommendations p 20 ln 429 should also mention Asia since Cambodia is in Asia

References p 23 ln 568 Ref 44 should put all author names in full, not abbreviations

Author Response

6/2/2022 Response to Reviewer Comments: -

Reviewer Comment 1: Abstract  p 1 ln 29 secondary 'education' reduced。

Author Response to Comment 1: Thank you for your suggestion. It has been updated.

Reviewer Comment 2: Fig 1 p 4 column Records Excluded should mention what was the reason for excluded

Author Response to Comment 2: Thank you for your feedback. The reasons have been enlisted. 14 studies were omitted as groups of caries and no- caries information was missing. 28 studies were omitted due to the lack of missing prevalence data.

Reviewer Comment 3: Data extraction p 5 ln 142 dmft/DMFT score  

Author Response to Comment 3: Thank you for your suggestion. It has been updated.

Reviewer Comment 4: Study Characteristics p 5 ln 188 (N=11,115)

Author Response to Comment 4: Thank you for your suggestion. It has been updated.

Reviewer Comment 5: Table 1 p 7 Peres 2005 what is HAZ?

Author Response to Comment 5: HAZ stands for “Height for age.” This has been updated in the table as well.

Reviewer Comment 6: Fig 2 p 8 forest plot depicting the Odd ratio on sugary drink/sweet consumption 0.1-0.5 should be 'Favours no caries' whereas 1-10 should be 'Favours caries' (similar to the one on good toothbrushing on the lowest bottom graph) i.e. Favours caries on the right side and Favours no caries on the left side

Author Response to Comment 6: Thank you for pointing that out. The figure has been replaced with the correct label and labeled too for your perusal.

Reviewer Comment 7: Fig 3 p 13 ln 264-265 it should mention which one defined A,B, C in the upper graph series, also there was errors defining the forest plot depicting OR  i.e. Favours caries on the right side and Favours no caries on the left side (of all 3 lower graphs)

Author Response to Comment 7: To the reviewer, placements have been made on the bottom right corner of all subgraphs. Please review all through A-F.

Reviewer Comment 8: Fig 5 p 15 also similar errors i.e. Favours caries on the right side and  Favours no caries on the left side (of all 4 graphs)

Author Response to Comment 8: Thank you for pointing that out. The figure has been replaced with the correct label and labeled too for your perusal.

Reviewer Comment 9: Limitations and recommendations p 20 ln 429 should also mention Asia since Cambodia is in Asia

Author Response to Comment 9: Thank you for your recommendation. It has been updated.

Reviewer Comment 10: References p 23 ln 568 Ref 44 should put all author names in full, not abbreviations

Author Response to Comment 10: Thank you for noting. It has been updated.